# Estimating the Heterogeneous Causal Effects of Parent–Child Relationships among Chinese Children with Oppositional Defiant Symptoms: A Machine Learning Approach

**DOI:** 10.3390/bs14060504

**Published:** 2024-06-18

**Authors:** Haiyan Zhou, Fengkai Han, Ruoxi Chen, Jiajin Huang, Jianhui Chen, Xiuyun Lin

**Affiliations:** 1Faculty of Information Technology, Beijing University of Technology, Beijing 100124, China; zhouhaiyan@bjut.edu.cn (H.Z.); mochen12134@emails.bjut.edu.cn (F.H.); jhuang@bjut.edu.cn (J.H.); chenjianhui@bjut.edu.cn (J.C.); 2Beijing International Collaboration Base on Brain Informatics and Wisdom Services, Beijing 100124, China; 3Engineering Research Center of Intelligent Perception and Autonomous Control, Ministry of Education, Beijing 100124, China; 4Engineering Research Center of Digital Community, Ministry of Education, Beijing 100124, China; 5Institute of Developmental Psychology, Faculty of Psychology, Beijing Normal University, Beijing 100875, China; chenruoxi@mail.bnu.edu.cn; 6Beijing Key Laboratory of Applied Experimental Psychology, National Demonstration Center for Experimental Psychology Education, Faculty of Psychology, Beijing Normal University, Beijing 100875, China

**Keywords:** oppositional defiant symptoms, multilevel family model, parent–child relationship, causal forest model, causal effect, heterogeneity

## Abstract

Oppositional defiant symptoms are some of the most common developmental symptoms in children and adolescents with and without oppositional defiant disorder. Research has addressed the close association of the parent–child relationship (PCR) with oppositional defiant symptoms. However, it is necessary to further investigate the underlying mechanism for forming targeted intervention strategies. By using a machine learning-based causal forest (CF) model, we investigated the heterogeneous causal effects of the PCR on oppositional defiant symptoms in children in Chinese elementary schools. Based on the PCR improvement in two consecutive years, 423 children were divided into improved and control groups. The assessment of oppositional defiant symptoms (AODS) in the second year was set as the dependent variable. Additionally, several factors based on the multilevel family model and the baseline AODS in the first year were included as covariates. Consistent with expectations, the CF model showed a significant causal effect between the PCR and oppositional defiant symptoms in the samples. Moreover, the causality exhibited heterogeneity. The causal effect was greater in those children with higher baseline AODS, a worse family atmosphere, and lower emotion regulation abilities in themselves or their parents. Conversely, the parenting style played a positive role in causality. These findings enhance our understanding of how the PCR contributes to the development of oppositional defiant symptoms conditioned by factors from a multilevel family system. The heterogeneous causality in the observation data, established using the machine learning approach, could be helpful in forming personalized family-oriented intervention strategies for children with oppositional defiant symptoms.

## 1. Introduction

Oppositional defiant disorder includes a variety of emotional and behavioral problems characterized by a recurrent pattern of angry/irritable moods, argumentative/defiant behavior, and vindictiveness toward authority figures [1,2,3]. As one of the most prevalent mental health disorders among children, the average prevalence rate is approximately 3–4% [1]. In China, the prevalence rate is believed to be 2.3–8% [4,5,6]. Moreover, based on the rating of parents or teachers [3,7,8,9], oppositional defiant symptoms can be found in a larger group of children and adolescents not directly diagnosed as having oppositional defiant disorder by professional psychiatrists or authoritative organizations [9,10]. Previous studies have identified numerous factors associated with oppositional defiant symptoms, whereas family factors, such as weak and poor familial contextual conditions [7], bad parental psychopathology [3,8,9,10], poor disciplinary practices [7], frequent interpersonal conflicts [10], and low emotional regularity ability [7,10] are known to be significantly associated with the development of oppositional defiant symptoms in children. However, for forming more effective intervention strategies, research is needed to further clarify the underlying pathways linking these factors to oppositional defiant symptoms. 

### 1.1. Multilevel Family Model and Oppositional Defiant Symptoms

The multilevel family model is proposed to explain the effects on the development and maintenance/exacerbation oppositional defiant symptoms in children across the entire family level, dyadic level, and individual level [9,10,11]. Factors at the entire family level include surface features (such as socioeconomic status) and deep features (such as family functions). Socioeconomic status is a comprehensive evaluation of a family’s economic income and social status, and family monthly income is a representative indicator. Many studies have shown that poor economic conditions, such as low family income and unemployment [12,13], are closely related to oppositional defiant symptoms. Family function refers to the interaction of physical, emotional, and psychological activities among all family members and is composed of family cohesion and adaptability. Considering the family as a complete unit and system, factors at the entire family level have proven to be closely related to destructive behavior in children, including oppositional defiant symptoms. Specifically, excessive conflict and contradictions in the family will worsen the relationship between family members and lead to destructive behavior in children [14,15,16,17], while family cohesion is a protective factor for children in terms of them developing oppositional defiant symptoms [18]. Moreover, factors at the entire family level can not only impact oppositional defiant symptoms directly but also indirectly through factors at the dyadic level (e.g., the relationship between the parent and children) and the individual level (e.g., the child’s emotion regulation abilities) [9]. 

The dyadic level, as a subsystem level, refers to the interactions between two or more family members, including a husband–wife subsystem and a parent–child subsystem. In the husband–wife subsystem, the marital relationship is one of the most impactful family risk factors in relation to children’s disruptive behaviors. Lower marital quality, such as more marital conflict and violence between intimate partners, is associated with more oppositional defiant symptoms in children [19,20]. In the parent–child subsystem, a large number of research has emphasized the strong association with oppositional defiant symptoms. On the one hand, children with oppositional defiant symptoms have more conflicts with their parents in comparison with children without oppositional defiant symptoms [21], and children with oppositional defiant symptoms are associated more with subsequent depression and less happiness, which can be influenced by the parental attachment [4]. On the other hand, a maladaptive parenting style is very critical in increasing oppositional defiant symptoms [6,8,15]. Children with oppositional defiant symptoms are more likely to experience maltreatment, including physical and emotional abuse and emotional neglect [22,23]. It is also found that factors at the dyadic level can have direct effects on oppositional defiant symptoms and indirect effects through individual characteristics [9]. 

The individual level mainly focuses on the individual family members themselves, including both the parent’s individual level and the child’s individual level. With respect to individual parent factors, parental individual characteristics (e.g., parental depression, aggression, anxiety), cognitive factors (e.g., parental negative attribution style) and emotion factors (e.g., emotion regulation) are included [8,24,25,26]. One of the main contributing factors to the development of oppositional defiant symptoms in children is parental emotion regulation. Research shows that parental emotion dysregulation was significantly and positively related to child’s oppositional defiant symptoms [22,27]. At the individual level of children, child emotion regulation has proven to be a consistently strong predictor of child emotional and behavioral problems [7,10]. The multilevel family model has proposed that factors at the individual level are the most proximal factors linked to oppositional defiant symptoms [9,11,22]. 

In summary, the multilevel family model respectively integrates the related family factors into the entire, dyadic and individual levels to increase understanding the development of oppositional defiant symptoms. Moreover, studies have demonstrated contributions from these factors to oppositional defiant symptoms. Many family factors could affect the development of oppositional defiant symptoms directly, but also, the factors highly interact and are associated with the oppositional defiant symptoms. It is still worthy of investigating the complicated linkages between these factors and oppositional defiant symptoms along with the family system perspective.

### 1.2. Parent–Child Relationship and Oppositional Defiant Symptoms

The parent–child relationship (PCR) is one of the most important factors at the dyadic level, and reflects the attention parents give to children and the attachment of children to parents [28]. Basically, a strong and secure PCR can provide a stable and supportive environment for children, which can mitigate the negative effects of family conflicts and be conducive to their healthy growth [29]. In contrast, a poor and impaired PCR can increase the risk of oppositional defiant symptoms. In a study with a large sample [7], the parents’ support showed a direct association with the oppositional defiant symptoms in preschool children. It is also suggested that the PCR plays a pivotal role in the development and exacerbation of oppositional defiant symptoms in Chinese children [6,10]. Moreover, many intervention strategies are focused on the management of PCR [6,30], and have demonstrated the feasibility and effectiveness of improvements in PCR across different cultural contexts [31]. Behavioral problems in young children, including oppositional defiant symptoms, could be alleviated by improving PCRs with behavioral intervention strategies [32]. They can also enhance parents’ parenting skills and sense of parenting efficacy [33]. However, despite the strong association, it is quite valuable to precisely predict the potential benefits of improvements in PCR in terms of oppositional defiant symptoms at the individual level or among different populations for providing personalized advice before implementing a targeted intervention strategy.

Additionally, as a two-way interpersonal relationship between parents and children, the impact of the PCR on oppositional defiant symptoms is quite complex. On the one hand, there are indirect associations the between the PCR and oppositional defiant symptoms mediated by other family factors. Firstly, the PCR could affect oppositional defiant symptoms through child emotional regulation [6,7]. Children with a poor PCR have more difficulties in regulating their emotions and behaviors, which leads to a greater risk of developing anxiety and other internalized problems [34]. Furthermore, the level of temperament-related negative affects and sensory regulation can further serve as mediators in the relationship between parent support and oppositional defiant symptoms [7]. Moreover, a longitudinal study further showed that the quality of the PCR in the first year could affect the risk of oppositional defiant symptoms in the third year through the level of parental depressive symptoms in the second year [5]. On the other hand, PCR can also serve as a mediator linking factors on the entire family level [4,11]. A family environment characterized by frequent conflicts and tense relationships may lead children to learn negative interpersonal patterns, including undesirable methods of emotional regulation and behavioral control, thereby increasing the risk of oppositional defiant symptoms [34]. Considering a specific dimension of the PCR, parental alienation plays a mediating role between family violence and oppositional defiant symptoms, which indicates that family violence increases children’s feeling of alienation from parents and that higher parental alienation contributes to more oppositional defiant symptoms [35]. Hence, there is a need to more deeply understand the mechanisms of the PCR in the development of child oppositional defiant symptoms and explain the complex relationship between them by considering the family factors.

### 1.3. Causal Inference and Its Heterogeneity

Causal inference based on observational data has received attention from scholars in mathematics and machine learning, which can hierarchically consists of three layers with different causal concepts for reasoning: association, intervention, and counterfactuals [36,37,38]. Association deals with purely ‘observational’ and factual information, which is what machine learning usually achieves by recognizing what is happening. For example, regression models [38] are often used to assess the relationship between variables. In addition to this, the method of using the structural equation model (SEM) could establish correlations when considering multiple dependent variables simultaneously [39]. These methods have been widely used in human behavior and psychology studies. 

The second layer of intervention is about the effect of the action, which encodes information about what would happen if a certain intervention is hypothesized. The typical question asked in child development is “What happens if we carry out the behavioral intervention strategy?”. Finally, counterfactuals deal with information about what would have happened if a certain intervention had been carried out, from a counterfactual perspective. The typical question asked is “How would the outcome be different now if the behaviour had been different in the past?”. A new data-driven approach, the causal forest (CF) model, has been proposed for explaining interventional causality as well as counterfactual causality with observational data [40,41,42,43]. 

The CF model comprises a number of causal trees consisting of a single root node and multiple child nodes with binary branches. First, the samples in the CF model are assigned into two groups according to the conditions of an independent variable. Then, every single causal tree is grown based on the principle of maximizing the variance in estimated causal effects in each node by comparing the outcomes or values of the dependent variable within the two groups in each node [43]. Meanwhile, all the covariates in each node are balanced between the two groups for controlling their effects on the causality estimation. Until the tree cannot be divided further, the causal effect estimated in the leaf nodes (the final nodes in the tree) serves as the individual-based causal effect or all the samples in both groups defined by independent variable. To decrease selection bias and increase robustness, the data samples and covariates are randomly selected to generate new trees and calculate new causal effects, iteratively forming the causal forest. Each sample may be allocated in a different node across the trees, and a weighted individual-based causal effect is obtained. Hence, based on the individual-based causal effects, the causal influence of an independent variable on the dependent variable is estimated without the traditional manipulation process of randomization. Thus, the CF model can explore interventional causality by estimating the individual-based causal effect between two groups as well as balancing the covariates in leaf nodes. Moreover, the CF model can further explore counterfactual causality via an estimation of the individual-based causal effect. Specifically, for the samples in both the groups defined by the independent variable, the counterfactual outcomes can be obtained by calculating the difference between the estimated individual-based causal effect and the factual outcomes. Actually, causal inference with observation data is a fundamental problem in a variety of domains, and the method of CF modeling has been widely used to evaluate causal effects, including policy evaluation [42], disease causality evaluation [44,45,46], and evaluations of the impact of psychological interventions on student achievement [41].

Moreover, the CF model divides samples into subgroups based on the differences in multiple covariates, which makes it possible to analyze the heterogeneity of causal effects. The heterogeneity of causal effects focuses on assessing different causal effects across individuals or subgroups. Individual-based causal inference in the CF model can predict the counterfactual results for each individual, e.g., what the treatment effect would be if the individuals in the control group received the treatment. This is very useful to guide personalized treatment strategies or the formation of public policies [46]. For instance, the causal effect of federal spending and its associated economic effects on decreasing crime was larger in below-median-income counties [47]. Preschool children in rural areas with less educated mothers benefited more from health insurance. The installation of cameras reduced the occurrence of traffic accidents in densely populated and poor areas more significantly.

### 1.4. Aims of the Study

In this study, to more deeply understand the underlying mechanism of the PCR in oppositional defiant symptoms and predict the potential benefits at the individual level or among different populations, we focused on the effect of the PCR on the development of oppositional defiant symptoms, to further infer the causal relationship and estimate the heterogeneity on observational data. To address the issue, we trained the CF model with follow-up ODD samples in China. The questions and hypotheses of interest are as follows: (1) Is there an “interventional” and “counterfactual” causality for the PCR in oppositional defiant symptoms with the machine learning-based approach of the CF model? Based on the strong association in previous research with the methods of SEM and intervention studies, we hypothesized that there would be a deeper causality between them via CF modeling. (2) Since many family factors were observed to link the PCR and oppositional defiant symptoms, what is the deeper mechanism of the PCR in the development of oppositional defiant symptoms and how can we explain the complex relationship between family factors? Guided by the multilevel family model and heterogeneous causality estimation in the CF model, we hypothesized that some related family factors from the entire, dyadic and individual family levels are involved in affecting causality, exhibiting the heterogeneity across individuals and subgroups. (3) Despite the strong association found in previous studies, testing it in these sample data as supplementary evidence is necessary. Therefore, the last goal is to further assess the “associational” level of causality and test the prediction of oppositional defiant symptoms for PCR. We hypothesized that improvements in the PCR could directly predict the development of oppositional defiant symptoms after controlling the related family factors with hierarchical multiple regression (HMR).

## 2. Materials and Methods

### 2.1. Participants

The samples came from a large-scale longitudinal research project on oppositional defiant disorder, and was taken from 14 elementary schools in northern (Beijing), eastern (Shandong Province) and southwestern (Yunnan Province) Mainland China during 2013 and 2014. First, invitation letters and informed consent forms were sent to the parents and children by school teachers to recruit the participants, then the parents and children with complete consent forms were involved in the subsequent investigation. The inclusion criteria were as follows: (1) Children were enrolled in the first grade to the six grade in the schools; (2) There was no obvious organic or neurological disease in the children. The children and parents who could not participate in the project during the follow-up period were excluded. The basic demographic information and assessments of oppositional defiant symptoms and other family factors of both children and parents were recorded and assessed in two consecutive years [4,10,11,14]. In total, there were 307 boys (62.1%) and 187 girls (37.8%) involved in this study. The children’s ages ranged from 6 to 13 years old (*M* = 9.37, *SD* = 1.59), and the parents’ ages ranged from 25 to 64 years old ($M_{mother}$ = 37.03, *SD* = 3.98; $M_{father}$ = 39.23, *SD* = 4.91). 

Prior to conducting the study, the research protocol was reviewed and approved by the Ethics Committee of Beijing University of Technology (No. xxxb202404-2) and the Institutional Review Board of Beijing Normal University (No. 202003310034 and 202302280035), respectively. All procedures performed in the studies were in accordance with the ethical standards of the institutional research committee and with the 1964 Helsinki Declaration and its later amendments or comparable ethical standards.

### 2.2. Measures

#### 2.2.1. Assessment of Oppositional Defiant Symptoms (AODS)

The children’s parents assessed their oppositional defiant symptoms on an 8-item scale according to the Diagnostic and Statistical Manual of Mental Disorders (DSM-IV-TR) [48] both in the first year and second year, named baseline AODS and outcome AODS, respectively. The severity of oppositional defiant symptoms was evaluated by adding together the scores of items, with a higher total score indicating more oppositional defiant symptoms. Cronbach’s α of the scale was 0.85 in this study. 

#### 2.2.2. PCR

The Chinese version of the Child-Parent Relationship Scale (CPRS) [49,50] was used to assess the PCR, which was calculated using the sum of two subscales: closeness and conflict. Closeness measured parents’ feelings of affection and open communication with their children (10 items; e.g., I share an affectionate, warm relationship with my child). Conflict measured parents’ conflict with their children (12 items; e.g., My child easily becomes angry at me). The higher the total summed score, the better and stronger the parent–child relationship. In this study, Cronbach’s α of the scale was 0.82.

#### 2.2.3. Monthly Income

A 5-point scale for monthly income was used to evaluate the entire family’s economic level. A higher score suggested higher monthly income in a family.

#### 2.2.4. Family Cohesion/Adaptability

The Family Adaptability and Cohesion Evaluation Scale (FACES-II) [51,52] was used to evaluate cohesion and adaptability in the family. The subscale of cohesion was used to evaluate the emotional connection between family members (16 items; e.g., At home, we do things together), and the subscale of adaptability examined the ability of a family to change in response to problems at different stages (14 items; e.g., Our family likes to try different ways to solve problems). The total score of the questionnaire was the sum of the two subscales scores. A higher score on the FACES suggested stronger adaptability and cohesion in the family. In this study, Cronbach’s α for FACES-II was 0.84. 

#### 2.2.5. Marital Relationship

The Dyadic Adjustment Scale (DAS) [53] was a 32-item questionnaire used to measure the perception of the relationship with an intimate partner (e.g., Do you confide in your mate). A higher score of DAS indicated a higher-quality marriage. Cronbach’s α for the DAS was 0.89 in this research.

#### 2.2.6. Parenting Style

The Authoritative Parenting Index (API) [54] consisted of two subscales: responsiveness and demandingness. The responsiveness subscale (API-r) was reported by the parent and reflected the support of parents for their children (7 items; e.g., I listen patiently to my child). The higher the score, the higher the degree of support from parents. Cronbach’s α for this subscale was 0.82. The demandingness subscale (API-d) reflected the control of parents over their children (9 items; e.g., I always tell my child what to do). The higher the score, the higher the degree of control from parents. In this research, Cronbach’s α for this subscale was 0.81.

#### 2.2.7. Parent Emotion Regulation 

The self-reported Difficulties in Emotion Regulation Scale (DERS) [55,56] was used to assess parents’ ability to engage in emotion regulation (36 items; e.g., When I am upset, I become angry with myself for feeling that way). A higher score on the DERS indicated greater difficulty in emotion regulation being experienced by the parents. Cronbach’s α was found to be 0.84 in this study.

#### 2.2.8. Child Emotion Regulation

The Emotion Regulation Checklist (ERC) [57] was a 24-item measure used to assess children’s positive and negative emotion-related behaviors as reported by parents (e.g., Is easily frustrated). A higher score based on the ERC indicated poorer emotion regulation in children. In this study, Cronbach’s α was 0.82.

### 2.3. CF Modeling

The procedure of CF modeling was implemented in R studio based on the GRF (Generalized Random Forests) package [58,59]. GRF is used as an R package that contains numerous models, such as causal forests, random forests, etc., which can provide corresponding methods and tools for the estimation of heterogeneous causal effects. For details on the package, please refer to its technical reference (https://github.com/grf-labs/grf accessed on 13 June 2022).

#### 2.3.1. Data Preprocessing

As shown in Figure 1, the independent variable, X, dependent variable, Y and covariates, Cs, were first defined for CF modeling. 

The independent variable, X, is usually a binary value used to divide samples into two groups, and is further used to generate causal trees based on the principle of maximizing the variance in estimated causal effects in each node by comparing the values of the dependent variable within the two groups. To achieve this goal, we first defined the X values according to the difference in CPRS scores (range from −20 to 36) between the two consecutive years and divided the samples into an improved group and a control group. As shown in Figure 2, the differences in CPRS scores in the two consecutive years in most of the samples were within 10%. To balance the maximization of the difference between the improved PCR group and the control group as well as to ensure the inclusion of as many samples as possible, the samples with improvement in CPRS scores (10% change) were defined as the improved group, while those with relative stable scores (within 10% change) were defined as the control group. Therefore, in total, there were 423 qualified samples selected for CF modeling, with 155 children (37%) in the improved group and 268 children (63%) in the control group. In total, there were 274 boys (64.8%) and 149 girls (35.2%) involved. The children’s ages ranged from 6 to 13 years old (*M* = 9.35, *SD* = 1.58), and the parents’ ages ranged from 25 to 64 years old ($M_{mother}$ = 36.12, *SD* = 3.75; $M_{father}$ = 38.26, *SD* = 4.63). 

For the outcome of oppositional defiant symptoms, the AODS score in the second year was defined as the dependent variable. The values of dependent variables were used to calculate the causal effects at the leaf nodes by comparing the differences between the improved PCR group and the control group.

The other measurements in the first year were used as Cs to evaluate their impacts on causality. The measurements at the entire family level (monthly income and FACES), dyadic level (DAS, API-r, and API-d measurements), and individual level (DERS and ERC) in the multilevel family model and the baseline AODS score in the first year were defined as Cs in the CF model. 

The expectation–maximization (EM) method was employed to handle missing values. This statistical approach uses a probabilistic model and iteratively updates parameters until convergence is achieved for estimating missing values. SPSS 26.0 was utilized.

#### 2.3.2. Model Construction

Because branch dividing in each node of the tree relies on the differences in covariates, the CF model may lead to over-sensitivity to heterogeneity. Hence, two regression forest models were first introduced to estimate the influences of the related family factors, Cs, on the causality of the PCR (X) and the outcome AODS (Y) (shown in Figure 1). Specifically, we put the previously defined Cs and Y into the random forest model to estimate the $\hat{Y}$. In the same way, we further put the Cs and X into the random forest model to estimate the $\hat{X}$. Then, all the parameters ($X,\;\hat{X},\;Y,\;\hat{Y},\;Cs$) were put into the causal forest to adjust the influences of Cs on the estimation of causal effect. 

Furthermore, the regular causal forests were used to estimate the causal effects and the differences across samples. As shown on the right in Figure 1, each causal tree was generated by fitting the data and characterized with binary branching in the root node, multiple leaf nodes, and other intermediate nodes. The trees were grown based on the principle of maximizing the variance in estimated causal effects in each node by considering Cs; therefore, the samples in the leaf nodes for each tree had similar values for Cs and the causal effect was estimated by comparing the outcome AODS (Y) in the groups with and without improved PCRs. Specifically, during the generation of causal trees, all the Cs were used for the division of the binary branch in each node. In this way, the influences of the covariates on the causality could be controlled to ensure a more rational estimation of the causal effect. Because each sample might have been assigned to different leaf nodes across trees, a weighted mean was calculated to obtain the individual-based causal effects for the samples during CF modeling. Hence, based on the individual-based causal effects, the causal influence of the PCR on the oppositional defiant symptoms in the second year was estimated without the traditional manipulation process of randomization. Furthermore, with the continuous branches being divided in the causal tree based on covariates, the heterogeneous causal effects of the PCR on oppositional defiant symptoms could be further estimated.

Moreover, the strategy of honest tree was adopted in the CF model to decrease overfitting. The training samples were divided into two sub-samples, with one used to generate the tree and the other used to estimate the causal effect within the leaf nodes. The optimal number of trees in the forest was adjusted to 300 to minimize the variance in causal effect. Other parameters were set to the package default.

#### 2.3.3. Model Evaluation

The CF model serves as an estimator for quantifying causal effects, thereby obviating the necessity for cross-validation with the division of the training set and testing set, which is typically employed in predictive analytics in conventional machine learning. Nonetheless, it remains imperative to examine reliability when estimating causal effect and heterogeneity to ensure a comprehensive understanding of the variability across the subsets of samples in each tree. A test with the best linear predictor (BLP) was performed to assess the goodness of fit of the CF model to the longitudinal samples. Furthermore, it was also able to evaluate whether or not the heterogeneity of the causal effect had been well calibrated [60]. With the BLP test, the predicted dependent variable could be estimated as follows in Formulas (1) and (2):(1)$$Y=b_{0}\left(C\right)\left(X=0\right)$$
(2)$$Y=b_{0}\left(C\right)+\beta_{1}\tau \left(C\right)+\beta_{2}∆\tau \left(X=1\right)$$

According to Formula (1) for the control group (X=0), the output result was the baseline value, $b_{0}\left(C\right)\;$. On the other hand, according to Formula (2), for the improved group (X=1), the output result included two more components. With the individual-based causal effects, the average causal effect, $\tau \left(C\right)$, reflected the overall difference between the improved and control groups. In addition, the heterogeneity ($∆\tau =\tau -\bar{\tau }$) reflected the fluctuations in the causal effect between individuals. Basically, a coefficient of 1 for $\beta_{1}$ reached the significant level, suggesting that the mean forest prediction was reasonable. Similarly, a coefficient of 1 for $\beta_{2}$ also reached the significant level, additionally suggesting that the heterogeneity across subgroups estimated from the forest was well calibrated. 

#### 2.3.4. Analysis of Causal Effect and Its Heterogeneity

Based on the CF modeling, the group-level causal effect could be further analyzed with the obtained individual-based causalities. The average causal effects (ACEs) for all samples, the improved group (ACI) and control group (ACC) and their 95% confidence intervals (CIs) were calculated. An independent two-sample *t*-test was used to compare the different causal effects between the improved group and control group.

The heterogeneity of the causal effect can typically be further analyzed in two ways. The first one is to compare the different causal effects between subgroups based on single Cs. Here, we divided the group into two subgroups according to the median value of each C, and then compared the differences between the high and low subgroups [14,42]. The second way is to analyze the heterogeneity based on a representative tree selected from the CF model, and was able to analyze the heterogeneous causal effects in several subgroups by considering the important Cs together [45].

### 2.4. Analysis of HMR 

The selected samples in CF modeling were involved in HMR analysis with SPSS 26.0. Similarly, the critical independent variable was defined as the change in the PCR, which was calculated based on the difference of CPRS scores between the two consecutive years. The AODS score in the second year was defined as a dependent variable. Finally, the measurements at the entire family level (monthly income and FACES), dyadic level (DAS, API-r, and API-d measurements), and individual level (DERS and ERC) in the multilevel family model as well as the baseline AODS score in the first year were defined as covariates. Therefore, during the analysis, all the covariates were first entered as predictors into the regression model, and then the change in the PCR was entered as a predictor into the second regression model. The F-test was used to assess the fitting of the regression model, and the *t*-test was used to assess the contributions of the covariates and the independent variable. 

## 3. Results

### 3.1. Descriptive Statistics

In the analysis of the *t*-test, there was no difference between the two groups for the baseline AODS, while the outcome of the AODS in the second year was different (t (421) = 2.571, *p* < 0.001) (Table 1), suggesting that oppositional defiant symptoms in the improved group had been alleviated more than they had been in the control group. The group difference was also observed in the API-d (t (421) = −1.131, *p* < 0.05) and ERC (t (421) = −2.843, *p* < 0.001). 

### 3.2. Causal Effect of PCR on Oppositional Defiant Symptoms

The results of the BLP test are shown in Table 2. The estimated $\beta_{1}$ was 0.97 (*p* < 0.001), suggesting the causal effect was well estimated. The ACE for all samples was estimated as −0.727, and the negative value indicated significant improvements in oppositional defiant symptoms with PCR improvement. Moreover, as expected, there was a larger causal effect in the improved group (t (421) = 3.093, *p* < 0.05) (Table 3), which further indicated the causal effect of an improved PCR on oppositional defiant symptoms. 

### 3.3. Heterogeneous Causal Effects

#### 3.3.1. General Analysis of Heterogeneity

The estimated value of $\beta_{2}$ was 1.37 (*p* < 0.001) (Table 2), indicating the reliable heterogeneity of the causality. Based on the weighted sum of how many times the Cs were split at each depth in the causal forest, we could analyze the contribution of the Cs to the heterogeneity prediction. The results are shown in Figure 3. The most important Cs were baseline ODD, on which the forest spent 37% of its split. Apart from that, the proportions of FACES, API-r, DERS, and ERC were higher than 10%.

#### 3.3.2. Heterogeneity among Cs

As shown in Figure 4A, both the monthly income and FACES at the entire family level affected the causal effects, which suggested that children in families with a low income, or those from families with poor cohesion and adaptability, saw stronger benefits from PCR improvement (both *p* < 0.05).

As for the dyadic level (Figure 4B), first the children in families with a lower DAS score had greater causal effects (*p* < 0.05). Then, based on the two subscales of the API, we observed a reverse pattern of heterogeneity. The children in families with high API-r or high API-d value showed greater causal effects (both *p* < 0.05), suggesting the parenting style played a positive role in the causal effect. 

The emotional regulation abilities of parents and children were crucial factors at the individual level. When parents and children had poorer emotional regulation abilities, the causality was greater (*p* < 0.001) (Figure 4C).

Finally, we also observed the role of this in heterogeneous causality based on the baseline AODS. Based on the division of subgroups according to the median value of the baseline AODS, the causal effect in the high-baseline AODS group was greater than that in the low-baseline AODS group (*p* < 0.001) (Figure 4D). This result further suggested that the causal effect was strongly dependent on the severity of oppositional defiant symptoms.

#### 3.3.3. Heterogeneity Based on Causal Tree

Based on the Cs ranking result, we selected trees including the following five variables: baseline AODS, FACES, API-r, DERS, and ERC. There were three trees matching the criteria. Then, we selected one tree with the greatest difference in causal effects on the root node. As shown in Figure 5, the tree divided the sample of children into two parts on the root node based on the baseline AODS, and the children in the right part had more severe oppositional defiant symptoms than those in the left part. Their causal effect was mainly influenced by the FACES score, and Subgroup 6 showed the greatest causality in the whole tree. The findings were consistent with the results based on the ranking of important Cs and the heterogeneity between single variables, indicating the key roles of the FACES score and the severity of oppositional defiant symptoms in the causal effect. Moreover, to further maximize the variance in the causal effect, the causal tree continued to be divided into two branches with selected covariates, and this process was repeated until the causal tree could not be further divided. There were more covariates involved in influencing the causal effects. The smallest causal effect was observed in Subgroup 3 in the whole tree, in which the children had relatively lower ERC, higher DERS and lower API-r scores, indicating that multiple Cs affected the causality jointly. 

### 3.4. Results of HMR Analysis

First, the VIF for all predictors included in the models ranged from 1.00 to 2.99, suggesting that the multicollinearity was not a concern in the study. Then, both the regression models significantly fit the data (both *p* < 0.001 with F(8, 422) = 52.893 in Model 1 and F(9, 422) = 60.698 in Model 2). Finally, the details of the regression models are listed in Table 4. After controlling the effects of other predictors, entering the change in the PCR significantly improved the explanation of the variance in Model 2 ($\Delta R^{2}$ = 0.064, *p* < 0.001), suggesting its unique contribution in predicting the oppositional defiant symptoms.

## 4. Discussion

In this study, we aimed to train a data-driven CF model and elucidate the causal relationship and its heterogeneity between PCR and oppositional defiant symptoms under the framework of the multilevel family model. The CF model showed a significant causal effect of the PCR on oppositional defiant symptoms, which provides further evidence of causality estimation with observational and longitudinal data. Furthermore, we found that the causal effect was varied in individuals or subgroups based on the heterogeneity analysis, suggesting factors from multiple levels in a family play different roles in causality. These results contribute to a deeper understanding of the mechanism of the PCR on the development of oppositional defiant symptoms and enrich the theory of the multilevel family model [6,9,11]. From the perspective of prevention and intervention, the discovery of heterogeneous causality could be extremely helpful when forming more targeted strategies.

Based on previous research with the methods of SEM and intervention studies, there may be a potential causal link between the PCR and oppositional defiant symptoms [9,22,34,61,62]. In the present study, we first observed a unique prediction of oppositional defiant symptoms from a PCR change with the method of HMR, which is consistent with the previous findings. Furthermore, a causal effect from the PCR on oppositional defiant symptoms was further clarified with the machine learning-based CF approach. During CF modeling, the causal trees were grown by maximizing the difference in causal effects between improved PCR group and control group in each node. At same time, the baseline AODS score and factors within the multilevel family model were accounted for to ensure comparability. Therefore, the approach of the CF model was very similar to the method of a randomized controlled trial (RCT), but the causality was estimated in naturally developing samples with no real intervention. To further eliminate the selection bias often found in intervention studies based on the RCT method, we repeated this process hundreds of times by randomly selecting samples and calculated individual-based causal effects with the weighted means in the CF model. Hence, by comparing it with SEM and HMR, the CF model was found to be able to uncover the causal relationships on the interventional and counterfactual levels rather than relying on association alone, and our results provide new evidence to demonstrate a deeper causal relationship between PCR and oppositional defiant symptoms in observational and longitudinal data.

Moreover, the prediction based on the CF model may be more practical in reality. Based on the methods with an associational causality reference, for a new sample, the development of oppositional defiant symptoms in the second year can be predicted using the difference in the PCR between the two consecutive years while controlling the influences from other related family factors in the first year, as observed from the results of the HMR. On the other hand, based on the interventional and counterfactual causality estimated with the CF model, just the PCR and other covariates in the first year are required to predict the oppositional defiant symptoms in the second year for a new sample, which provides the opportunity to make predictions in advance with time and cost savings. Hence, the findings with CF modeling may be very helpful in evaluating the potential benefits from the improvement in the PCR in terms of oppositional defiant symptoms at the individual level before implementing the targeted intervention strategy. 

By considering the theoretical support for the multilevel family model, three factors are addressed. First, as many family factors contribute to the development of oppositional defiant symptoms [4,5,6,7,8,9,10,11], guided by the multilevel level family model, we can investigate the causal relationship between PCR and oppositional defiant symptoms by integrating all the factors from the entire, dyadic and individual family levels, along with the family system perspective of oppositional defiant symptoms. Second, consistent with the multilevel level family model [6,9,10,11], the findings for the direct association between PCR and oppositional defiant symptoms are further supported by the causal effect estimated via CF modeling and the prediction effect determined in the HMR analysis when family factors were taken as covariates in the present study. Finally, knowledge about the interactions between the family factors in the multilevel level family model is enriched in present study, which is mainly related to the heterogeneous effect of PCRs on oppositional defiant symptoms. The multilevel family model has illustrated the mediating role of the dyadic family level between the entire level and individual level in linking to oppositional defiant symptoms [4,7,9,10,11,35], whereas the present study exhibits the moderating roles of the family levels in the association of PCRs and oppositional defiant symptoms. FACES scores on the entire family level, API-r scores on the dyadic level, and the DERS and ERC scores on the individual family level were the most important factors contributing to the causal relationship between the PCR and oppositional defiant symptoms. Moreover, the heterogeneity can be further revealed by considering multiple family factors together with an illustration of a representative causal tree. The heterogeneous causal effects across individuals and subgroups indicate quite a complicated mechanism of the PCR in oppositional defiant symptoms, which can be divided into approximately three types.

The first one is related to the expectation of or support for oppositional defiant symptom alleviation. The baseline AODS and FACES scores were the most two important Cs and used as the root node and secondary node, respectively, in the representative tree. Notably, the effect of the baseline AODS score did not reflect the longitudinal change in the comparison between the baseline AODS and the outcome AODS scores as in intervention studies, while it was used as one covariate to estimate the influence of the causal effect during CF modeling. The multilevel family model proposes that factors at the entire family level play a basic supporting function in the family system [22], and are distal factors that affect oppositional defiant symptoms indirectly via more proximal factors, such as the PCR and child emotion regulation [16,22,62]. When children with oppositional defiant symptoms have more emotional and behavioral problems, there are more likely to be conflicts with their parents [4,9]. In such families, family members were less supported by each other [63]. Therefore, it might be more urgent for children with severe oppositional defiant symptoms and a less harmonious family atmosphere to alleviate oppositional defiant symptoms by improving the PCR. 

The second type is related to the ethological or manipulable explanations for oppositional defiant symptoms. Children with lower emotion regulation abilities themselves or had parents with lower emotional regulation abilities could benefit more by reducing oppositional defiant symptoms via PCR improvement. The results advance the knowledge of the effects from individual family factors on the development of oppositional defiant symptoms. Research has emphasized that parents’ negative emotional handling styles may easily impair children’s emotional regulation skills, and subsequently lead to an increase in the risk of oppositional defiant symptoms [11,64]. Furthermore, a high-quality PCR could not only have a unilateral effect on a child’s development but also can influence the parents’ relationship and functioning [22]. Based on the multilevel family model, we further propose factors at the individual level might affect the causality of the PCR in oppositional defiant symptoms.

The last type of heterogeneity is related to the parenting style, particularly the factor API-r, which plays a compensatory role. The parenting style is one factor from the dyadic level of the parent–child subsystem. Generally, positive parenting styles play protective roles in the development of oppositional defiant symptoms [65]. It is suggested that factors at the dyadic level may directly or indirectly have an effect on oppositional defiant symptoms by mediating/linking the entire family-level factors and individual-level parent and children factors, so the interaction of family factors between levels exhibits a hierarchical connection [22]. However, the influence of factor interplay within the family levels in oppositional defiant symptoms is rarely considered in research. The observed compensatory role of parenting styles on the causality in this study provides preliminary knowledge about the mutual link in the parent–child subsystem at the dyadic level. Parenting style is highly related to the PCR, and emphasizes parents’ leading role in behavior control and manipulation. To some extent, the PCR is a consequence of parenting style. A reasonable explanation for the larger causal effect in children with higher API-r scores is that parents can actively respond to their children’s needs and, accordingly, the children’s oppositional defiant symptoms can be alleviated significantly. 

There are some limitations to this study that should be considered in the future. The first is related to the data samples. The samples used in this study were selected from an ordinary school population, and oppositional defiant symptoms were assessed by the parents of the children. Although this method is commonly used when investigating behavioral problems in children from a community [3,7,8,9,14], it is still very limited in terms of allowing us to generalize conclusions to the samples from clinical institutions. Moreover, the findings in this study need to be verified with larger sample sizes. The second limitation is related to the methodology. For some considerations, we set a criterion based on the difference between the two consecutive years for the binary group division, but it was still subjective and more validations are needed. Furthermore, in the CF model, the causal effect is estimated by comparing the group difference in each node with matching Cs. This group-based method may also still have selection bias. Currently, many “real” individual-based methods are being developed to predict counterfactual outcomes and are aimed at reducing the influence of the binary definition of the individual variable [66], so they could be introduced in future research. Finally, we mainly focused on the psychological factors under the framework of the multilevel family model to estimate the heterogeneous causality in this study, while research has found that the occurrence and development of oppositional defiant symptoms varies with the factors gender, age and social culture [2], so whether or not the causality effect would be different across the subgroups based on these factors is still not clear. 

Despite these limitations, the following section briefly outlines some important implications for the prevention/intervention of oppositional defiant symptoms from both theoretical and practical perspectives. First, the findings from the current study highlight the importance of the management of the PCR in a family. Research has shown that there may be a reverse link, with severe oppositional defiant symptoms leading a to bad PCR [9], though we did not evaluate the causality in this direction. This is because based on a consideration of the practice, the effects of improvements in the PCR on alleviating oppositional defiant symptoms are of more interest. Second, the application of a machine learning-based CF model offers a novel research technique in investigating the causal relationship between PCR and oppositional defiant symptoms in observation data. Meanwhile, this approach can quantitatively represent the variability in this causal effect between different individuals or groups in detail, which may help in evaluating the intervention strategies aimed at more rational PCR management. 

## 5. Conclusions

A machine learning-based CF model was introduced to investigate the causal relationship between PCR and oppositional defiant symptoms with observational and longitudinal data. The causality and heterogeneity across individuals and subgroups could be helpful in more deeply understanding the mechanism of PCR on the development of oppositional defiant symptoms as well as in forming precision- and personalization-based prevention/intervention strategies. 

## Figures and Tables

**Figure 1 behavsci-14-00504-f001:**
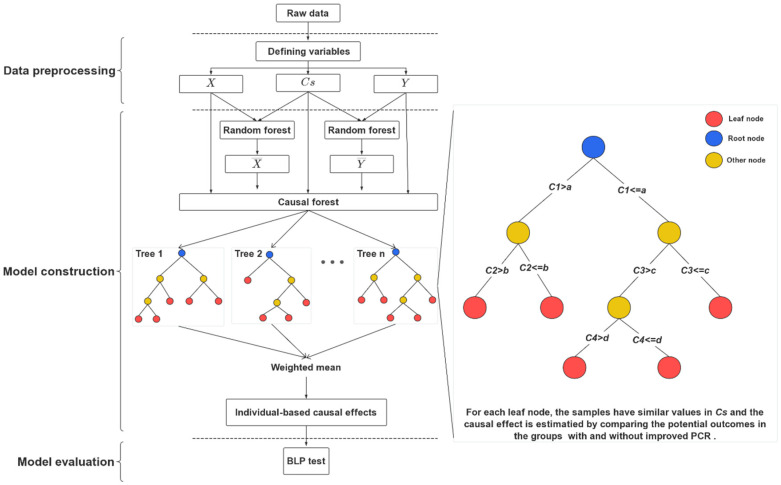
Overview of causal effect estimation with CF modeling.

**Figure 2 behavsci-14-00504-f002:**
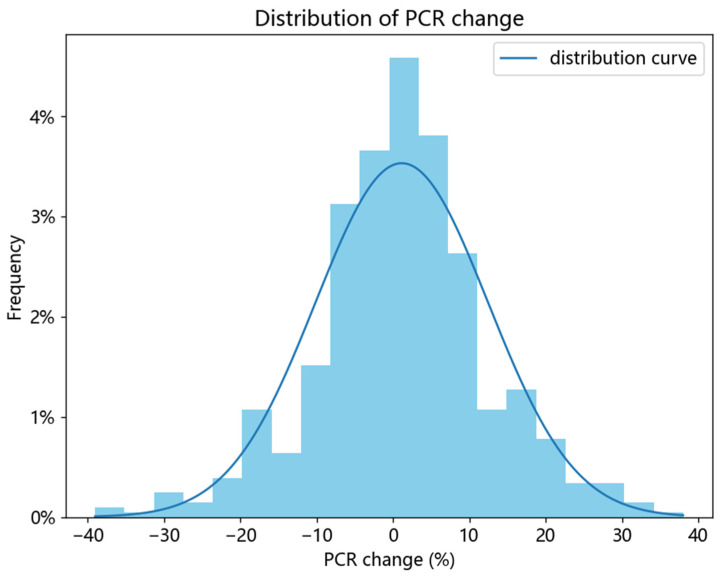
Histogram of the distribution of PCR changes in the two consecutive years.

**Figure 3 behavsci-14-00504-f003:**
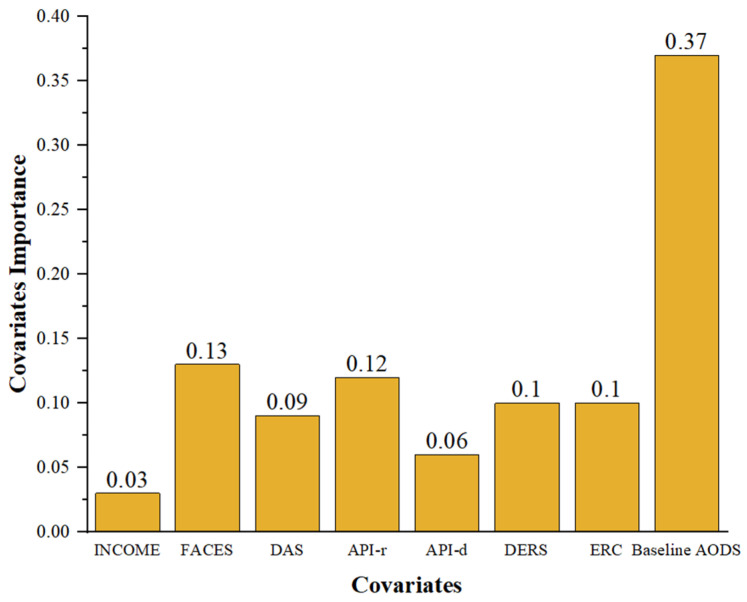
The ranking of importance of Cs.

**Figure 4 behavsci-14-00504-f004:**
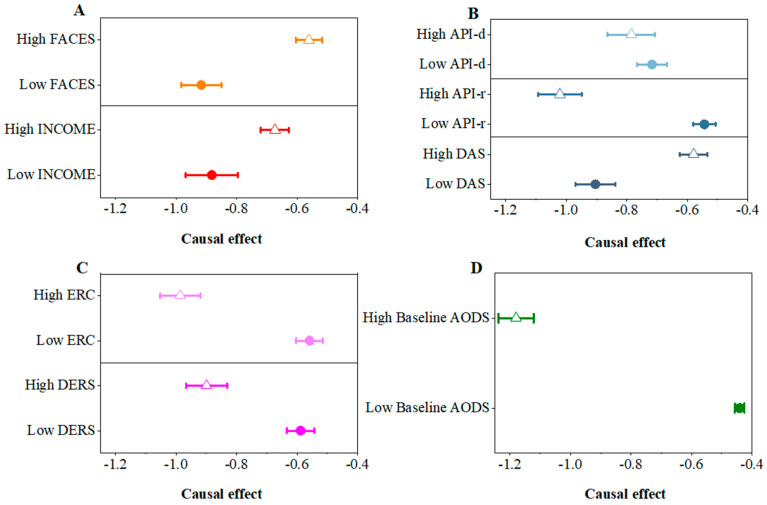
Heterogeneity along single covariates. The influence on the causality of PCR on the oppositional defiant symptoms from the factors in the entire family level (**A**), dyadic level (**B**), individual level (**C**) and the baseline AODS (**D**), respectively. The empty triangels represent the groups with higher values of the scales and the solid circles represent the groups with lower values.

**Figure 5 behavsci-14-00504-f005:**
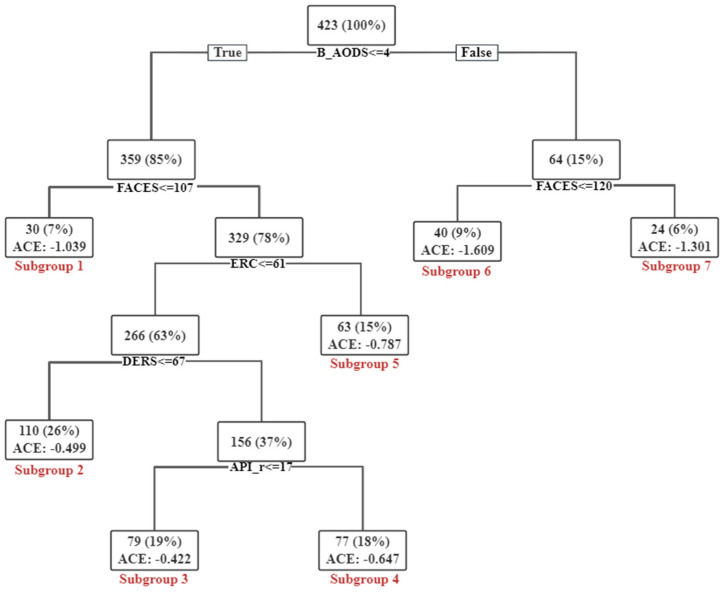
Subgroups identified by the representative causal tree. ACE: average causal effects.

**Table 1 behavsci-14-00504-t001:** Summary in improved group and control group.

	Improved	Control	t	*p*
	Mean/SD	Mean/SD		
Outcome AODS	0.94/1.40	1.39/1.93	2.571	<0.001
Baseline AODS	2.12/2.29	1.62/2.23	−2.215	0.276
Income	2.87/0.87	3.07/1.05	1.970	0.066
FACES	121.97/14.18	124.94/15.69	1.941	0.734
DAS	134.63/15.08	135.13/17.70	0.294	0.755
API-r	18.96/3.75	17.71/3.63	−3.367	0.444
API-d	13.88/3.50	13.51/3.03	−1.131	<0.05
DERS	76.64/14.07	72.98/14.19	−2.564	0.639
ERC	56.94/11.35	52.41/17.88	−2.834	<0.001

**Table 2 behavsci-14-00504-t002:** BLP tests for causal effect and heterogeneity.

	Estimated	S.E.	*p*-Value
$\beta_{1}$	0.94	0.16	<0.001
$\beta_{2}$	1.37	0.36	<0.001

Note. S.E. standard error.

**Table 3 behavsci-14-00504-t003:** Causal effect obtained from causal forest.

	Estimated	95% CI
ACE	−0.727	(−0.974, −0.480)
ACI	−0.854	(−0.881, −0.827)
ACC	−0.653	(−0.892, −0.414)

Note: ACE: average causal effect for the whole data; ACI: average causal effect in the improved group; ACC: average causal effect in the control group; CI, confidence interval.

**Table 4 behavsci-14-00504-t004:** Results of HMR analysis.

	Outcome AODS	
Predictors	$\Delta R^{2}$	Beta
Model l	0.505 ***	
Baseline AODS		0.609 ***
Income		−0.035
FACES		0.010
DAS		−0.244 ***
API-r		0.031
API-d		−0.116 **
DERS		−0.011
ERC		−0.048
Model 2	0.064 ***	
Baseline AODS		0.623 ***
Income		−0.042
FACES		−0.019
DAS		−0.193 ***
API-r		0.052
API-d		−0.094 **
DERS		0.002
ERC		0.002
Change in PCR		−0.266 ***

Note. level of significance: ** *p* < 0.01, *** *p* < 0.001.

## Data Availability

The datasets generated during and/or analyzed during the current study are available from the corresponding author on reasonable request.

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
