# Peer review of "Estimating the Heterogeneous Causal Effects of Parent–Child Relationships among Chinese Children with Oppositional Defiant Symptoms: A Machine Learning Approach"

_behavsci, 2024, doi:10.3390/bs14060504_

Round 1

Reviewer 1 Report

Comments and Suggestions for Authors

Comments on the Quality of English Language

Minor revision is needed and some spelling errors are required to be corrected.

Reviewer 2 Report

Comments and Suggestions for Authors

Congratulations to the authors for conducting this research. Observations made on the article are:

- The contextualization of the research problem and the support of its importance are well done in the introduction.

- The need for this study is well argued, with evidence from the literature

- The objectives of the study are well written.

- In the "participants" section adding the number given by the review committee to the research protocol would be recommended.

- The instruments used to assess the variables are correctly presented.

- Results are presented in a comprehensive way

- The discussion section is also well structured and formulated.

- The bibliography consists mainly of recent studies.

Reviewer 3 Report

Comments and Suggestions for Authors

This study investigated the causal effects of parent-child relationship on oppositional defiant disorder (ODD) symptoms by using a machine learning based causal forest  model.

My first major concern is the concept of “ODD symptoms.” Although the ODD criteria on the DSM-IV-TR were used in this study, the results were not equal to “ODD symptoms.”  These symptoms can be found in children and adolescents without ODD. Therefore, I would like to suggest the authors to use “oppositional defiant symptoms” instead of “ODD symptoms.”

The second major concern is the reason for use oppositional defiant symptoms as the outcome variable and parent-child relationship as the predictive variable. Although environmental factors may contribute to the development of oppositional defiant symptoms, harsh, inconsistent, or neglectful child-rearing practices play an important role in the causal theories of ODD. I would like to suggest the authors to test the prediction of oppositional defiant symptoms for parent-child relationship.

Minor concerns:

Subtitles: 1.2 …1.1…

Line 81: “undisirable” error?

Line 83: Kerns, 2013)..

The first paragraph of 1.3. was unrelated to this study.

Line 186: DSM-IVR: it should be revised into “DSM-IV-TR.”

Reviewer 4 Report

Comments and Suggestions for Authors

Brief summary: In this interesting study, the authors investigated the heteregeneity of parent-child relationship on children suffering from oppositional defiant disorder. 423 children were split into an ''improved'' and control groups and a machine learning model was deployed to assess different parameters in attempt to model parent-child relationships in regards to ODD symptoms. The authors found that causal effect was greater in children with higher baseline of ODD, difficult family system, lower emotion regulation abilities in themselves (child with ODD) or their parents. They conclude that these findings could help in forming personalized family-oriented intervention strategies for patients diagnosed with ODD. Please find my comment below.

iThenticate: The iThenticate expressed a 70% match with other content from the web (59% with a pre-print by the same authors on  :https://www.researchsquare.com/article/rs-2850159/v1)

Introduction:

- While the authors define ODD in the introductory sentence (lines 38-40), it it suggested to adopt either the DSM-5 or the ICD definition of ODD.

- Please list some of the factors suggested by the multiple studies listed from lines 44-46.

- Lines 48-54 are unclear: the line 48 mentions a subsystem, but no system has been previously mentioned or refered to. Please rephrase to account for clarity.

- Same goes for line 55 ''the individual level'' : from which conceptual model or which approach? A level belong to a model or a conceptual entity. Please clarify.

- Lines 125-127 : while reading Wang et al., 2022, the author does not mention anywhere causality. It is important to emphasize the distinction between causality and correlation. In machine learning models, it is very rare to establish causality. Using coefficients of independent variables in regards to a dependant variables will indicate, to some degree, a correlation between the independant variable and the dependant variable, but does not assess for causality. Same thing goes for line 134: no where does Athey & Wager state any form of causality with their approaches. Please clarify.

- The aim could be further precised and the authors could provide their hypotheses.

Overall the introduction is well constructed, could be shortened, but the structure is adequate and introduces the readership well to the problematic being assessed.

Materials and methods:

- How were participants recruited?

- Please state inclusion/exclusion criteria.

- Rational behind the use of GRF rather than other models should be briefly discussed. 

- How was the model validated? Usually cross-validation is conducted to account for reliability. BLP is unusual compared to other methods such as K-Fold.

- Considering the sample size, isn't there a risk of overfitting of the model? If so, how is this accounted for?

- How is statistical significance evaluated? Please briefly add a sentence to mention what is considered significant in your study.

Results:

- Even if patients characteristics are accessible in other studies, the authors are encouraged to provide a summary of these characteristics as they are essential to have a good appreciation of the population targeted by their work. 

- Overall results are well presented. Figure 3 could be enhanced for clarity.

Discussion:

- The discussion is well aligned with the current state of the literature.

- A conclusion section could be added to bonify the summary and put an emphasis on the relevance of the work conducted by the authors.

Minor comment: 

- Parent-child relationship (PCR) on line 63 does not have the same size as the rest of the manuscript. Same goes for observation and observation data in the subsequent paragraphs.

- The reference list states 61 references but the manuscript has 62 references (reference 61 has 2 references).

- It would be helpful to use sub-headings in the Materials and methods rather than plain text for the different sections.

Comments on the Quality of English Language

Nil

Round 2

Reviewer 1 Report

Comments and Suggestions for Authors

Overall, I appreciated that the authors revised the article to make it more explicit.  However, I still have a few questions that the authors may clarify further.

Please clarify the relationships among ODD, ODS, and children's disruptive behaviors, I still felt confused while reading the first paragraph of the article.

In line 46, what is “pf children”?

In line 48, how ODS can be facilitated?

All families’ factors listed in lines 49-51 were proved to be important to children’s disruptive behaviors, so this means all these factors also can impact ODS?

I am still confused about the research procedures of the current study.  As the authors mentioned in the cover letter, there is no intervention was conducted.  Then how to understand the independent variable X that is described in lines 299-308?

Why 10% change in CPRS score is appropriate to divide subgroups?

How do you divide the subgroups based on different covariates?

If there is no intervention was conducted, how to understand the baseline and the outcome? 

What are different functions of causal tree and HML to answer your research question and test your hypotheses?

Comments on the Quality of English Language

Minor editing of English language required.

Reviewer 3 Report

Comments and Suggestions for Authors

The authors have revised their manuscript based on the reviewer's suggestions. I would like to suggest the editors accepting it for publication.

Reviewer 4 Report

Comments and Suggestions for Authors

The reviewers responded to all my previous queries. I have no further comments.

Comments on the Quality of English Language

Nil
